# Effects of Perceived Traffic Risks, Noise, and Exhaust Smells on Bicyclist Behaviour: An Economic Evaluation

**Stefan Gössling** [1,2,3,*], **Andreas Humpe** [4], **Todd Litman** [5] **and Daniel Metzler** [4]

1   Western Norway Research Institute, P.O. Box 163, 6851 Sogndal, Norway
2   Department of Service Management and Service Studies, Lund University, Box 882,
    251 08 Helsingborg, Sweden
3   School of Business and Economics, Linnaeus University, 391 82 Kalmar, Sweden
4   Department of Tourism, University of Applied Sciences, 80636 Munich, Germany;
    andreas.humpe@hm.edu (A.H.); daniel.metzler@hm.edu (D.M.)
5   Victoria Transport Policy Institute, 1250 Rudlin Street, Victoria, BC V8V 3R7, Canada; litman@vtpi.org
*   Correspondence: sgo@vestforsk.no or stefan.gossling@ism.lu.se or stefan.gossling@lnu.se

**Abstract:** Active mode (walking, bicycling, and their variants) users are exposed to various negative externalities from motor vehicle traffic, including injury risks, noise, and air pollutants. This directly harms the users of these modes and discourages their use, creating a self-reinforcing cycle of less active travel, more motorized travel, and more harmful effects. These impacts are widely recognized but seldom quantified. This study evaluates these impacts and their consequences by measuring the additional distances that bicyclists travel in order to avoid roads with heavy motor vehicle traffic, based on a sample of German-Austrian bicycle organization members (n = 491), and monetizes the incremental costs. The results indicate that survey respondents cycle an average 6.4% longer distances to avoid traffic impacts, including injury risks, air, and noise pollution. Using standard monetization methods, these detours are estimated to impose private costs of at least €0.24/cycle-km, plus increased external costs when travellers shift from non-motorized to motorized modes. Conventional transport planning tends to overlook these impacts, resulting in overinvestment in roadway expansions and underinvestments in other types of transport improvements, including sidewalks, crosswalks, bikelanes, paths, traffic calming, and speed reductions. These insights should have importance for transport planning and economics.

**Keywords:** air pollution; cost-benefit analysis; cycling; Detours; exhaust fumes; transport externalities

---

## 1. Introduction

Active transport modes (walking, bicycling, and their variants, such as scooter and wheelchair travel) play unique and important roles in an efficient and equitable urban transportation system, including basic mobility for non-drivers, access to public transport, healthy exercise, and enjoyment. In most cities, 10–30% of total trips are made by these modes (Figure 1).

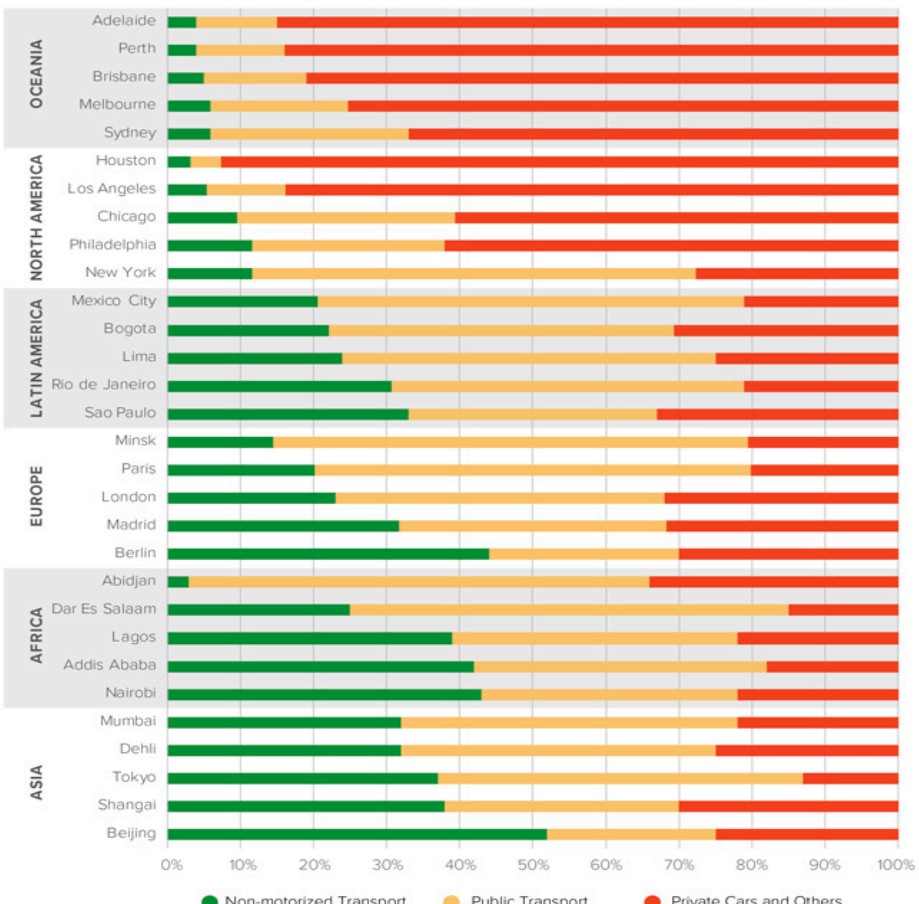

**Figure 1.** Active mode shares in selected cities. Source: [1].

Active modes can provide many economic, social, and environmental benefits, particularly when they substitute for motorized travel (Table 1). As a result, many jurisdictions have policies and programs to improve walking and bicycling conditions and encourage their use [2].

**Table 1.** Active transport mode benefits.

| Improved Active Transportation Conditions | More Active Transport Activity | Reduced Automobile Travel | More Compact Communities |
|---|---|---|---|
| <ul><li>Improved user convenience and comfort</li><li>Improved accessibility for non-drivers, which supports equity objectives</li><li>Option value</li><li>Higher property values</li><li>More neighborhood security</li></ul> | <ul><li>User enjoyment</li><li>Improved public fitness and health</li><li>More local economic activity</li><li>Increased community cohesion (positive interactions among neighbors) which tends to increase local security</li></ul> | <ul><li>Reduced traffic congestion</li><li>Road and parking facility cost savings</li><li>Consumer savings</li><li>Reduced chauffeuring burdens</li><li>Increased traffic safety</li><li>Energy conservation</li><li>Pollution reductions</li><li>Economic development</li></ul> | <ul><li>Improved accessibility, particularly for non-drivers</li><li>Transport cost savings</li><li>Reduced sprawl costs</li><li>Open space preservation</li><li>More livable communities</li><li>Higher property values</li><li>Increased security</li></ul> |

Source: [3].

Motor vehicle occupants are enclosed in vehicles, which provides comfort and protection. In contrast, active mode users are vulnerable to various risks and discomforts, including potential injuries, noise, and air pollution. A major obstacle to achieving active transport goals is what planners call the "barrier effect" or "severance", which refers to the disamenities that motor vehicle traffic imposes on pedestrians and bicyclists, including risks and pollution exposure [4]. This harms active mode users directly and discourages use of these modes, creating a self-reinforcing cycle of less active travel, more motorized travel, and more harmful effects.

These impacts are widely recognized but seldom quantified. This is a specific example of various negative externalities of motor vehicle travel [4–7]. Comprehensive analysis of these impacts is essential for accurate and fair evaluation of transportation policies and projects. For example, numerous studies have quantified and monetized (measured in monetary values) the costs of traffic congestion and the benefits of roadway expansions that could reduce congestion delays, but these studies generally ignore the barrier effect, that is, the additional disincentives and delays that wider roads and increased vehicle traffic impose on pedestrians and bicyclists. This omission exaggerates the predicted value of roadway expansion and underestimates the potential benefits of alternative congestion reduction strategies such as public transit service improvements and congestion pricing [8]. This suggests that, for more comprehensive and equitable planning, transportation practitioners (planners, economists, and policy analysts) need information to quantify and monetize the incremental delay and costs that motorized transport imposes on active modes.

This study helps to fill in this gap. It surveyed cycling organization members to determine the additional travel distances that they incur to avoid riding on high volume urban streets in order to reduce their exposure to motor vehicle traffic risk, noise, and air pollution. Standard travel cost values are used to monetize this additional travel, similar to methods that are commonly used to monetize travel detour and traffic congestion costs. This research should be useful to practitioners that are involved in multimodal transport planning and comprehensive policy analysis, and to bicycle advocates who want to support more balanced urban transport planning and investment practices.

## 2. Background

### 2.1. Bicyclist Exposure to Negative Externalities

When pedestrians and bicyclists travel on or near roads with heavy motor vehicle traffic, or want to cross such roads, they are exposed to various negative externalities, including exposure to accident risk, noise, and air pollution [9,10]. These impacts can be significant. For example, 27% of the 1.2 million road fatalities per year involve cyclists and pedestrians [11–14], and cyclists sometimes also experience harassment and intentional assaults by motorists [15,16]. These risks and threats tend to discourage bicycle use, particularly by non-regular riders [17,18].

Bicyclists are also exposed to noise, as they share road space with motorized vehicles or use tracks that are adjacent to roads. This can have significant health impacts [19,20]. Noise causes stress reactions that are linked to tinnitus, mood changes, chronic sleep disturbance, lack of recovery from tiredness, nervousness, and anxiety [20–25]. Various studies have shown that cyclists are exposed to high levels of traffic noise [26], which results in higher stress levels [9]. Noise is thus an annoyance for cyclists [27], as well as a potential health risk.

Motor vehicle air pollution is a serious health threat, particularly for people who are physically active near major roadways [2]. Motor vehicle traffic emits many harmful substances, including carbon monoxide (CO), nitrogen oxides ($NO_x$), hydrocarbons (HCs), particulate matter (PM2.5, PM10), and ozone [27–30]. Harmful health impacts include bronchitis and asthma, lung cancer and cardiopulmonary diseases, as well as heightened cardiopulmonary mortality [31–34]. The Lancet Commission estimated that air pollution, to which transportation makes a significant contribution, is responsible for 16% of deaths worldwide [2]; in Europe, it has been concluded that air pollution is responsible for 6% of total mortality, half of this attributed to motorized transport [29]. Active mode

users tend to experience particularly large air pollution health risks, since physical activity increases respiration rates and therefore pollutant absorption [26,35].

Overall, this indicates that motor vehicle traffic imposes significant negative externalities on active mode users, implying a cost.

*2.2. The Cost of Exposure*

The European Union relies on cost-benefit analyses (CBA) to determine the advantages and disadvantages of transport investments [36]. CBA is widely used to justify infrastructure developments, to assign infrastructure priorities, or to favour investments in specific transport modes [37]. As a decision-making tool, CBA has various weaknesses, such as the choice of parameters included in assessments, or the limited availability of data on included items (for discussion see e.g., [38–41]. CBA often focuses on one transport mode, and so it often overlooks impacts on other modes [8,42]. Hence, CBA will often consider the benefits of roadway improvements but overlook negative impacts that wider roads and more motor vehicle travel have on active modes, and since most public transit trips include walking and bicycling links, resulting in reductions in public transit access.

In the European Union, CBA is a standard assessment tool for transport investments, relying on the European Commission's Handbook on Transport Costs [36]. The Handbook recommends the inclusion of various externalities in all transport CBA, such as climate change, air pollution, noise pollution, vehicle operation, travel time, and injuries. This list is not comprehensive, and it omits some costs that motorized transportation imposes on other road users, such as traffic risks, congestion, health impacts, or discomfort [8]. More comprehensive CBA frameworks, such as those that are presented by the Victoria Transportation Policy Institute [4], include as many as 20 different economic impacts [8].

This study focuses on two issues that have received limited attention in transport cost–benefit analyses. First, there is evidence that bicyclists will detour around busy streets, dangerous crossings, and poor infrastructure to avoid exposure to traffic risks, pollution exposure, and other inconveniences that are caused by motorized traffic [10,43,44]. From a CBA-perspective, such detours represent a cost that is imposed on bicyclists by motor vehicle travel. In the existing literature [44], effects have not been disaggregated by traffic impacts (injury risk, noise, or air pollution). This study attempts to address this, with particular attention being paid to unpleasant traffic smells [27], which cyclists may seek to avoid.

## 3. Method

A combined qualitative and quantitative approach was used to address the research questions, i.e., the importance of exhaust vis-à-vis other traffic externalities (traffic risks and noise). In a first qualitative step, a sample of 20 German-speaking cyclists was identified through purposive sampling strategy (snowball sampling; [45]), and interviewed in semi-structured interviews in January and February 2018. This strategy was chosen because the population of cyclists is ultimately unknown, and as a qualitative approach on a so far unconsidered issue, such as exposure to vehicle exhaust, any group of cyclists represents an appropriate starting point for the understanding of perceptions. Interviews were initiated with students, and then expanded to a range of other cyclists. Interviews lasted 10–30 min, and they had the purpose of understanding general attitudes to exhaust fumes and other aspects of traffic potentially interacting with perceptions of smell. Interviews began with a broad question: "How do you see motorized transportation from your perspective as a cyclist?", and continued to discuss various aspects of traffic. Where respondents did not bring up exhaust fumes, they were asked specifically whether they reacted to smells in traffic.

Results of these qualitative interviews revealed that for some cyclists, exhaust fumes were a 'given', self-evident aspect of the contemporary transport system, though respondents reported to avoid trucks and their diesel fumes. For other cyclists, exhaust represented a nuisance, but it was not considered a health risk. Some worried about their health enough to wear protective masks when riding in heavy traffic. Strategies to avoid exhaust smells also included breathing through clothes,

covering one's nose/mouth with a hand, holding one's breath, to bypass vehicles with high exhaust levels (specifically busses, trucks, and motorcycles), or timing road crossings to minimize exposure. At red lights, cyclists reported to stay back from accelerating car fleets, or to get ahead of traffic to avoid exhaust fumes. Some travellers avoided bicycling under potentially bad traffic situations (e.g., rush hour), resulting in less total bicycling, particularly by children. Interviews consequently confirmed the importance of exhaust in transport behaviour, revealing a wide range of avoidance strategies.

Interviews also revealed variations in bicyclists' sensitivity to pollution. For instance, smokers appeared to be less worried about exhaust while parents with infants expressed greater concern. To avoid exhaust, cyclists reported taking longer routes that avoid heavy traffic streets, though such detours could be motivated by various reasons besides avoiding air pollution, such as reducing injury risks and noise or to enjoy more scenic routes. Route choice also depended on day time, as some roads were considered as 'no go' areas after dark; or situation-dependent, for instance, when being late for a meeting.

Results also indicated that perceptions of these impacts varied and may change over time. For example, traffic impacts in German cities, though far from ideal for cyclists, were perceived as better than in other countries. Cyclists also expressed an awareness of exhaust, possibly as a result of public discussions of diesel emissions and court orders restricting old diesel car models in cities. In summary, interviews confirmed the central importance for exhaust in cyclist transport behaviour. These insights were used to design the quantitative part of the study.

To collect data on cyclist transport behaviour and exhaust implications, a cooperation was initiated with the German Cyclists' Association (Allgemeiner Deutscher Fahrradclub, ADFC) as well as the Austrian Verkehrsclub Österreich (VCÖ). A questionnaire was developed and made available electronically, using the data collection platform Survey Monkey. The link to the questionnaire was sent to 9906 Twitter followers of the ADFC on 16 February 2018, as well as to the 1877 Facebook followers of the Austrian VCÖ, on 24 March 2018. VCÖ also highlighted the survey in their 22 March 2018 newsletter. A total of 491 valid responses were recorded in the week following the calls, n = 257 through the ADFC and n = 234 through the VCÖ. As the sample is self-selected and not representative of cyclists, or, indeed ADFC or VCÖ members, the results need to be seen as indicative. It is possible that the questionnaire had specific appeal to those most exposed to externalities of the motorized transport system, and caution needs to be exercised in generalizing its results.

The questionnaire was introduced in a general way, i.e., "Are cyclists influenced by traffic noise and exhaust fumes", followed by information that the survey was organized by researchers from Lund University, Sweden. Anonymity was assured. Questions addressed how many km respondents travelled per week. Data was collected for work/educational trips and leisure, with the latter comprising all other trips. Cyclists were asked how they viewed the importance of traffic risks, exhaust, and noise (rating based on 1–5 Likert scale). Estimates of detours as a result of risks, exhaust, and noise, i.e., the three factors that together determine the perception of road situations, were recorded in km and minutes of travel. As data was collected on the basis of estimates, this allowed for a comparison of distances and travel time, and hence the calculation of speed to assess the credibility of the data. Finally, the questionnaire focused on exhaust fumes, using a willingness-to-pay vis-à-vis willingness-to-accept approach, to economically determine the cost of exhaust. As qualitative interviews revealed, exhaust fumes from the combustion of diesel, petrol, oil, or other fuels are perceived as unpleasant in comparison to a traffic situation without such smells. Cyclists may place a premium on the avoidance of such smells.

Willingness-to-pay (WTP) refers to the amount of money that cyclists reported to be prepared to pay for non-exposure to exhaust fumes; willingness-to-accept (WTA) to the amount of money considered fair compensation for exposure to exhaust fumes. These are measures of consumer surplus, either as compensating gain (WTP) or compensating loss (WTA) [46]. Economic values were collected in Euro per week; the questionnaire recorded WTA before WTP. Note that, in economic studies, WTP usually yields lower values than WTA [47]. Yet, the EC Handbook on Transport Cost-Benefit

Analysis suggests that WTA "is the appropriate measurement for damage compensation analysis" (EC Handbook update). To calculate the cost of exhaust, two different measures are considered, i.e., avoidance behaviours (detours, non-use of bicycle), resulting in a time cost to cyclists [6,48] as well as WTA as a measure of the cost arising from the existence of exhaust. Both of the values are broken down to per km values.

It should be noted that since two datasets were combined, a non-parametric test for independent samples (Mann-Whitney-U test) was made, confirming that no statistically significant differences exist between the means of the two datasets (derived through ADFC/VCÖ), and that these could be combined and analyzed as one. Furthermore, a few adjustments were made to prepare the dataset. Regarding the distances cycled, one outlier (6000 km/week) was removed, as well as three responses where both the distance cycled for work and leisure was '0'. Detours registered as '0' in either km or minutes cycled were not included in the analysis of detours. In comparison, WTA and WTP analysis considered all of the responses where positive answers (including '0') were reported.

## 4. Results

### 4.1. Cycling Patterns and Detours

Respondents reported to cycle a total of 41,208 km/week, out of this 19,639 km for leisure and 21,569 km commuting to work. Averaged over the sample (n = 491), this corresponds to an average of 43.9 km/week for work, and 40.0 km/week for leisure, with an overall total average of 83.9 km/week (Table 2). There was considerable variation in bicyclists' behaviour. For example, some cyclists reported to not cycle to work at all, while others never cycled for leisure. Reported total cycle distances per cyclist also varied from between 1 km/week to 600 km/week. In comparison to other research, the averaged sample may be considered indicative of a more active cyclist population, though the results are not directly comparable due to the "per week" approach in this study (cf. [49–51]).

**Table 2.** Cycling: Overview of results.

| Distance (km/Week) | Work (km/Week) | Leisure (km/Week) | Total (km/Week) |
|---|---|---|---|
| Minimum | <1 | <1 | 1 |
| Maximum | 350.0 | 500.0 | 600.0 |
| Mean | 43.9 | 40.0 | 83.9 |
| Median | 30.0 | 30.0 | 70.0 |

Figure 2 shows the distribution of weekly distances, by distance class, for trips to work. As expected, most of the trips are short, following a distance-decay relationship; this is, the longer the distances, the lower the share of cyclists. Some 25% of respondents stated to cycle up to 10 km/week for either leisure or work. The peak in the distance class 91–100 km can be interpreted as an accumulation of estimates, i.e., where respondents did not know exactly how much they cycled, opting for a round-figure estimate. At the high end of the distance classes, a small share of very active cyclists reported high work and leisure related levels of cycling, though there is possibly also an accumulation of estimates at 150 km. The distance class of >150 km aggregates a group of very active cyclists.

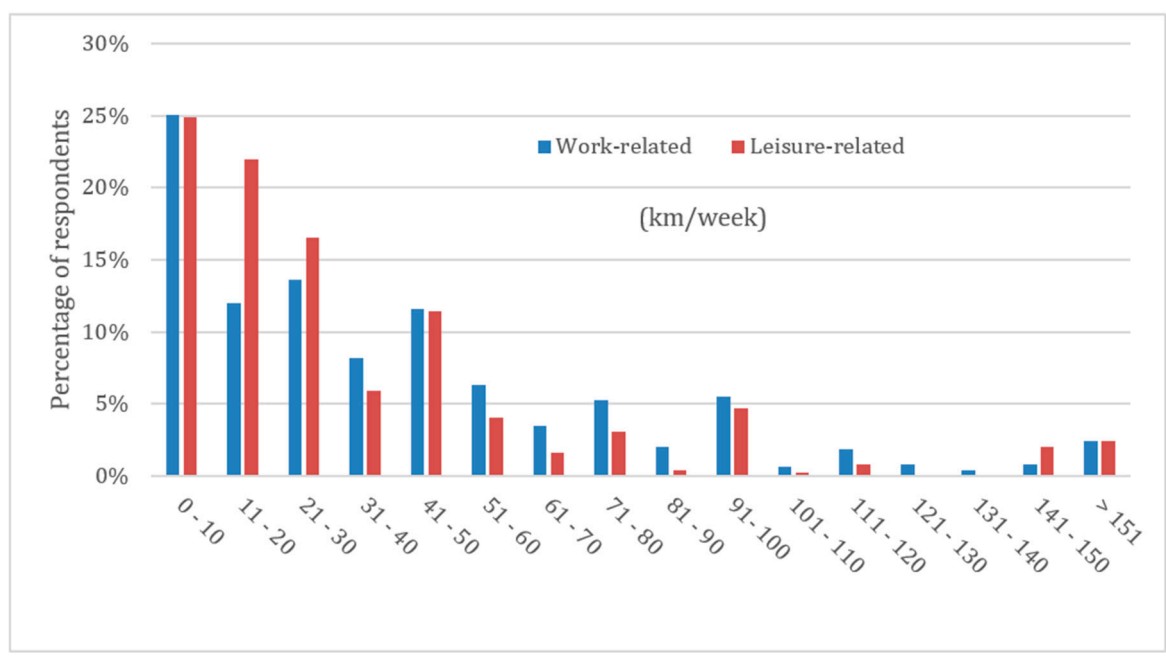

**Figure 2.** Work-/leisure-related cycling, km per week.

About three-quarters of respondents (n = 367; 74.7%) reported to cycle detours to avoid traffic risks, noise or exhaust fumes, with a maximum of 60 km/week or 180 min/week. The average detour cycled per week was 7.2 km (Table 3). For all respondents claiming detours, these amounted to 2656 km/week, or 7893 min. These figures appear to be internally consistent, as they translate into 2.97 min per km, or an average speed of about 20.2 km/h. The Danish Transportministeriet [52] estimates, in comparison, that the average cycle speed in Denmark is 16 km/h, but this comprises a far greater and diversified cyclist population. The detour distance (2656 km/week) as the share of the overall distance cycled (41,208 km/week) suggests that, if integrated over the whole sample of cyclists (n = 491), the average detour in this survey is 6.4%, to avoid traffic risks, exhaust, or traffic noise.

More specifically, a large share of cyclists (39%; n = 367) reported small detours, with up to 2.5 km/week, and another 28% of between 2.6 and 5.0 km/week (Figure 3). On the other end of the spectrum, five percent of cyclists claimed to cycle more than 20 km/week in detours. These results can be compared with detour times: Here, 52% (n = 354) reported to cycle up to 10 min in detours every week, with only a small percentage (6%) reporting more than one hour in additional distances cycled to avoid traffic risks, exhaust, or noise.

**Table 3.** Detours cycled.

| Distance (km/Week) | Total (km/Week) | Total (Minutes/Week) |
| --- | --- | --- |
| Maximum | 60.0 | 180.0 |
| Mean | 7.24 | 22.3 |
| Median | 5.0 | 10.0 |

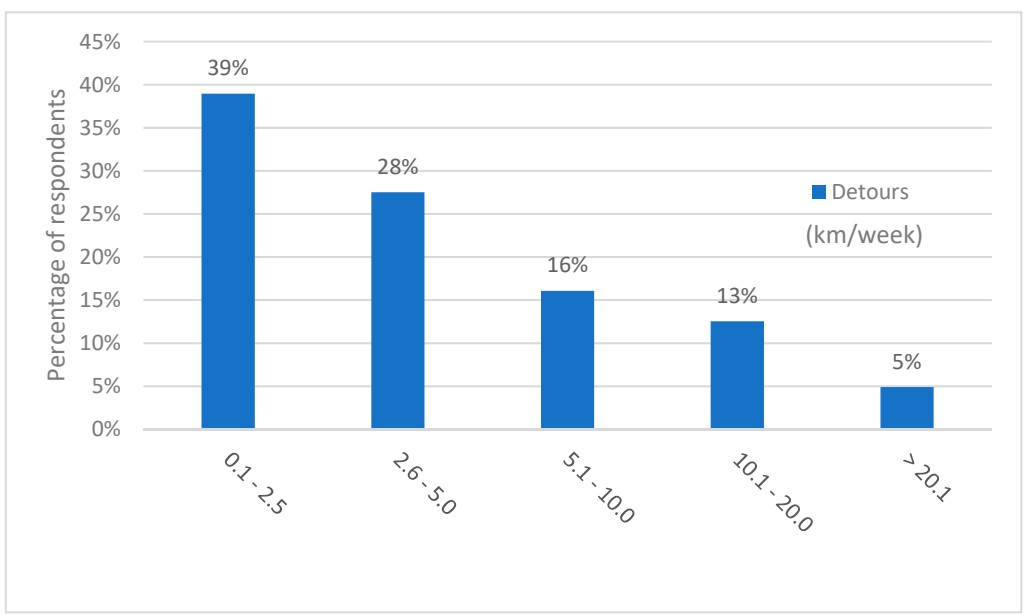

**Figure 3.** Detours cycled to avoid traffic risks, exhaust or noise.

*4.2. Importance of Exposure*

To better understand the relative importance of traffic risks, exhaust fumes, or noise for cyclists, respondents were asked to rate these three interrelated aspects on a Likert-scale from one, "very important", to five, "not important at all". As Table 4 shows, traffic risks are the main reason for detours, with 60% of respondents (n = 410) rating this item "very important", and another 31% as "important". This confirms research suggesting that the perception of traffic risks—which is different from real traffic risks—is important for cyclist behaviour [10,53]. Only 2% of cyclists rated traffic risks as less important, and none as not important at all. Exhaust is the second most important factor affecting cyclists, with 38% of respondents (n = 412) rating this item very important and another 43% as important. Again, only a small share of respondents rated exhaust as less important (6%) or not important all (1%). Noise shows a somewhat different distribution, with one-third of respondents (34%) rating noise as important and another 30% as less important. Clearly, noise is the least relevant of the three aspects. This is also evident from average ratings between 1–5, where risks rank highest over the entire sample at 1.5, followed by exhaust at 1.9, and noise at 2.8.

**Table 4.** Rating of noise, traffic risks, and exhaust.

| Rated as | Noise (n = 413) | Traffic Risks (n = 410) | Exhaust (n = 412) |
|---|---|---|---|
| Very important | 13% | 60% | 38% |
| Important | 34% | 31% | 43% |
| Neither important nor unimportant | 15% | 6% | 12% |
| Less important | 30% | 2% | 6% |
| Not important at all | 8% | 0% | 1% |
| Total | 100% | 100% | 100% |

*4.3. Cost of Detours*

These detours represent additional travel time by bicyclists to avoid externalities. Transport planning often seeks to minimize travel time [36,54]. Various studies around the world have monetized the value of travel time and travel time savings for economic evaluation purposes ([4,7,52]). Travel time is generally considered as a private cost that can be measured based on travellers' willingness to pay, though this value is context dependent. For example, under favourable conditions, cycling may have a positive value, due to enjoyment and fitness benefits, while under unpleasant

conditions, such as wet and cold weather along a busy roadway, costs can be high. In Denmark, where travel time estimates have been conducted since 2007 (Value of time study 2007), the time of cyclists is valued at an average €12.37/h for unrestricted cycling and €18.69/h for delays (current estimate according to Center for Transport Analytics [55]). At an average speed of 16 km/h for bicyclists, a travel cost of €0.79/km is the current basis (in 2017) for calculations in Denmark. Far lower values have been reported for Germany, where [48] suggest a value of time of €4.66/h, resulting in a cost of about €0.3/km for cyclists (without consideration of leisure/work trip purposes). When considering this latter value, the cost of time "lost" to detours of 7.24 km/week, multiplied with €0.30, i.e., about €2.17/week for the average cyclist in this survey, and referring to average cycling conditions. Note that this cost involves traffic risks, exhaust, and noise.

The cost of exhaust fumes, which also represents a private cost, can be estimated on the basis of WTP/WTA assessments. A total number of n = 307 respondents reported WTA and n = 355 WTP. Reported values are presented in classes, because the WTA dataset included a high number of very high values of up to €10 million. Such values are not realistic assumptions of WTA, but they mirror substantial irritation over exhaust exposure on the side of a significant number of cyclists. All the values higher than €100 were summarized in one class. The median WTP is €10/week, while the median WTA is twice as high, at €20/week. It is notable that in the lower range of up to €10/week, WTP appears to be higher than WTA (Figure 4). This distribution remains the same, even when all 0 values are removed. However, only in 26 cases (8.5% of sample) is the stated WTP actually higher than WTA. A potential explanation is that for some respondents, accepting that traffic exhaust is different from a 'no exhaust fumes' situation, i.e., the two scenarios are incommensurable for these cyclists, who place a greater value on clean air than being compensated for polluted air.

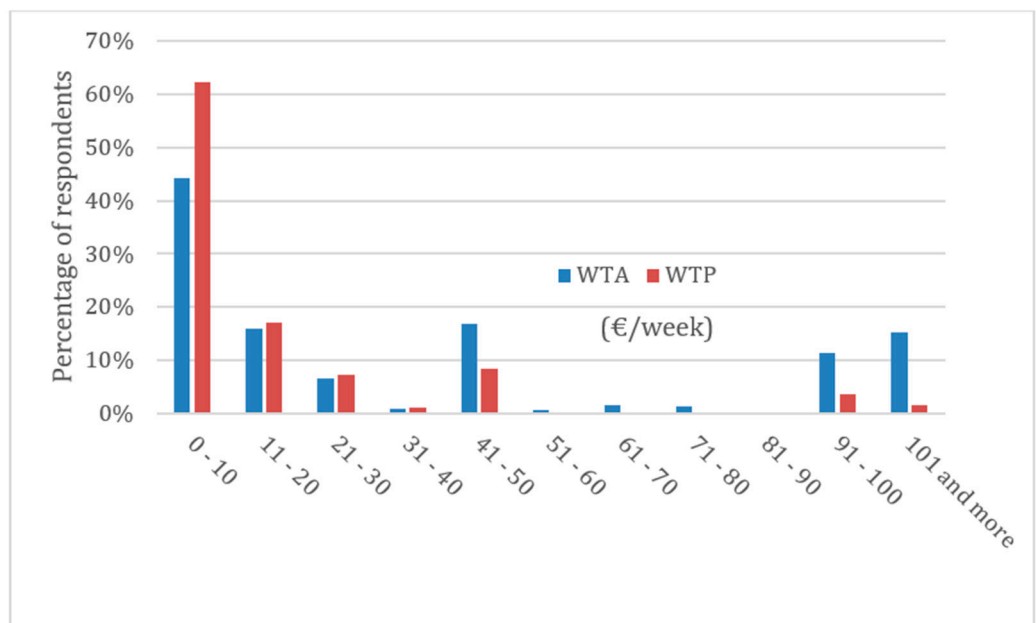

**Figure 4.** Willingness to Accept (WTA) and Willingness to Pay (WTP) for avoided exhaust fumes.

Table 5 shows that Likert ratings of the perception of exhaust fumes as a negative impact on cyclists is correlated with WTA and WTP. In the table, the average amount of money placed on WTA as well as WTP declines with the importance of exhaust as a traffic nuisance. Cyclists that considered exhaust very important also reported high WTA (€54/week) and WTP values (€23.5/week); these declined to €5/week and €4/week for those not considering exhaust important.

**Table 5.** WTA and WTP values placed on exhaust fumes, depending on perception.

| Likert Rating | Very Important | Important | Neither Nor | Less Important | Not Important at All |
|---|---|---|---|---|---|
| Exhaust WTA, average €/week | 53.93 | 36.45 | 28.84 | 24.16 | 5.00 |
| Exhaust WTP, average €/week | 23.53 | 16.69 | 12.93 | 12.44 | 4.00 |

To understand the statistical dependence between WTA/WTP and traffic risks in comparison to exhaust and noise, the Pearson and Spearman rank correlation coefficients were determined (Table 6). The calculation is based on metric WTA/WTP values and ordinal Likert-scale values for noise, traffic risks, and exhaust. WTA correlations are negative, as expected, and significant at $\alpha = 0.05$ for noise (Pearson and Spearman) and traffic risks (Pearson), as well as significant at $\alpha = 0.01$ for exhaust (Pearson and Spearman). This highlights the importance of exhaust as a barrier effect that has so far not received much attention as a negative externality in transport CBA.

**Table 6.** Correlation coefficients.

| Pearson Correlation | Noise | Traffic Risks | Exhaust |
|---|---|---|---|
| WTA | −0.12 ** | −0.11 ** | −0.26 *** |
| WTP | 0.00 | −0.04 | −0.09 * |
| **Spearman Rank-Order Correlation** | **Noise** | **Traffic Risks** | **Exhaust** |
| WTA | −0.11 ** | −0.09 | −0.25 *** |
| WTP | −0.04 | −0.03 | −0.09 * |

Statistical significance at 10% *, 5% ** and 1% ***.

As recommended by the European Commission [56], WTA values are used for the assessment of the cost of exhaust. Given the high number of outliers, the median value of €20/week is used for calculations, translating to an average cost of €0.24/km (with an average cycling distance of about 84 km/week). The private cost of exhaust exposure is thus significant, specifically if adding WTA (€0.24/km) and travel time invested in detours (€0.30/km). While results are not representative of the wider cyclist population, they indicate that exhaust is a major nuisance, a health risk, and a potential detriment to cyclists, an insight that should bear repercussions for transport CBA not currently considering this aspect.

## 5. Discussion

Before discussing the survey results, its limitations should be considered. As outlined, the survey represents a sample of cyclist and sustainable transport organisation members (the German ADFC, as well as the Austrian VCÖ). Furthermore, the questionnaire was distributed in February/March 2018, at a time when cold temperatures may have influenced perceptions, as exhaust fumes may have been perceived differently than during other times of the year. As a result, survey respondents are likely to have more than average riding experience and skill and so are not necessarily representative of the wider population of cyclists in these countries (cf. [49–51]).

To compare the results with national transport studies, the German mobility panel [57] reports that, for the average German, the average distance travelled is 41.2 km per day, out of this 71% by car, 20% by public transport, 3% by walking, 3% by bicycle, and another 3% falling on different transport modes (data for 2016). For cycling, this corresponds to about 1.25 km per day, which can be compared to the 12 km per day reported by cyclists in this study. When comparing travel times, differences become somewhat smaller, as KIT [57] reports an average cycling time of 6.5 min per day, as compared to 36 min in this study. The latter comparison would also suggest different cycling speeds

in the general German population (11.5 km/h) and in this study (20 km/h), perhaps a reasonable result, since the survey focused on cycling organization members who are probably experienced riders.

Against this background, the study reveals several relevant insights with relevance for transport economics and urban planning. First, traffic conditions significantly affect cycling patterns: Respondents in this study cycled on average 6.4% longer distances to avoid traffic risks, noise, and exhaust pollution. Results confirm that cyclists avoid specific roads, favouring safer, cleaner, or less noisy detours. A share of cycling is probably discouraged [10,53]. Where no alternatives for detours exist, or where these would be too time-consuming, cycling may not be an option, causing travellers to use other transport modes. Traffic planners should consequently devote more attention to reducing cyclists' exposure to noise and air pollution as well as accident risk. The results suggest that all three impacts contribute to these detours. Although most practitioners recognize the need to build cycling tracks that are separated from roads for safety sake, this study indicates the value of creating networks of dedicated bicycle streets totally separated from high-traffic roadways in order to provide options for risk-, exhaust-, and noise-free cycling.

A second major question relates to the incremental costs of these route detours. Motorized transport causes significant negative externalities, including costs that are imposed on users of other modes [8,14]. Although not necessarily representative of all cyclists, this study indicates that, even among experienced bicyclists, motor vehicle exhaust imposes €0.24 per km cycled, a significant cost compared with other transportation impacts. These costs are probably much higher for less experienced cyclists who are likely to take even longer detours or simply forego bicycling and shift to less desirable modes to avoid motor vehicle traffic impacts [58]. Such shifts directly harm individual travellers, who experience more stress and discomfort or are forced to use less preferred travel modes, and by reducing bicycling and increasing motor vehicle travel, these shifts also harm communities with increased traffic and parking congestion, accident risk, and pollution emissions imposed on society.

With a growing understanding of motor vehicle air pollution harms, cyclists may increasingly perceive exhaust as a significant health threat as well as a discomfort. This will further increase the costs of detours and bicycle travel foregone. Interviews that are conducted as part of this study indicate that parents with small children are particularly likely to refrain from bicycling for the fear of traffic risks or health impacts. As exhaust is primarily linked to trucks, buses, and motorcycles, policy makers and city planners also need to consider ways to reduce their negative impacts, such as limiting when and where they may operate and encouraging shifts to electric propulsion.

## 6. Conclusions

This study investigates the impacts that motor vehicle traffic imposes on non-motorized travel Although widely recognized by practitioners and the general public, these impacts are seldom quantified or measured in monetary units.

The study surveyed a group of bicycle and environmental organization members to determine how they respond to high levels of injury risk, noise, and air pollution on busy urban roadways. The results indicate that these cyclists often travel significantly longer distances to avoid motor vehicle traffic negative externalities. These detours caused survey respondents to ride 6.4% farther, on average, to reach their destinations. Since most survey respondents are relatively experienced bicyclists, the impacts on average cyclists is probably much greater. This suggests that motor vehicle traffic impacts impose significant incremental costs, including additional travel time, foregone bicycling, plus shifts to motorized travel, which increases both user and external costs.

The study used both Willingness to Pay (WTP) and Willingness to Accept (WTA), contingent valuation methods to estimate the value that cyclists place on avoiding traffic risks, exhaust, and noise. Averaged over the entire sample, and using the WTA median as basis for the estimate, the study indicates that incremental time costs average €0.24/km, a significant value when compared with other transport costs. The results also indicate that these negative externalities reduce total cycling travel. This creates a self-reinforcing cycle of less active travel, more motorized travel, and more

harmful external costs. These impacts tend to be the greatest for young and inexperienced bicyclists, and in urban areas. With growing concerns about air pollution health risks, these impacts are likely to increase in the future.

These results should have important implications for transport policy and planning analysis. To be efficient and equitable, economic analysis must be comprehensive, accounting for all the significant costs and benefits, including indirect and external impacts. Although the barrier effect (the disamenities that motor vehicle traffic imposes on non-motorized mode users) has been documented and quantified for decades, it is ignored in most transport cost benefit analyses (CBA). This biases policy and planning decisions to favour motorized over non-motorized modes, resulting in an overinvestment in roadway expansions and underinvestment in other types of transport improvements, such as better sidewalks, crosswalks, cycle tracks, paths, traffic calming and speed control, and transportation demand management strategies that reduce total motor vehicle traffic, such as public transit improvements and more efficient transport pricing. It also undervalues policies and planning decisions that reduce negative traffic externalities, such as replacing noisy and stinky diesel vehicles with electric powered vehicles.

Despite these biases, many jurisdictions are committed to improving and encouraging active transport, and reducing motor vehicle traffic, particularly in urban areas. Policy makers, practitioners, and the general public understand intuitively that active transport provides significant benefits, while motorized travel imposes significant costs that are overlooked in conventional economic evaluation. However, efforts to support active transport could be strengthened, and made more efficient and equitable, by incorporating impacts, such as the barrier effects of wider roads and the health benefits of more active travel, into formal policy and planning analysis.

This study had several limitations that future research can address. Surveys should include more diverse people including less experienced cyclists, people who want to bicycle but are deterred by unpleasant roadway conditions, and users of other active modes, such as walking, scooter, and wheelchairs. Research may also further disentangle the relative importance of traffic risks, exhaust, noise and other factors in influencing detours. This includes the question as to whether exhaust perceptions refer to smell as a nuisance, or whether smell is considered as a proxy for health impacts; in the latter case, it can be assumed that fear of cycling will increase with debates on the effects of harmful substances, such as $NO_x$ or PM2.5. Finally, it would be important to know how perceptions differ between quiet neighbourhood roads and highly polluted traffic corridors, and how this affects WTA values for these.

It would also be useful to investigate how frequently travellers forego active modes due to motor vehicle traffic impacts, and therefore how much walking and bicycling would increase, and motorized travel would decline, if a community improved walking and bicycling facilities, or reduced traffic noise and pollution, for example, by converting diesel buses and trucks to electric propulsion.

Practitioners need practical tools for quantifying and monetizing barrier effects so they can estimate the costs that wider roads and increased motor vehicle travel impose on active travellers, and the savings and benefits that would result from improving active mode travel conditions.

**Author Contributions:** Conceptualization, methodology, Writing-Original Draft Preparation, S.G.; Formal Analysis, A.H.; Writing-Review & Editing, T.L.; Conceptualisation, D.M. H.J. helped with data collection.

**Funding:** This research received no external funding.

**Conflicts of Interest:** The authors declare no conflict of interest.

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
