# Peer review of "Effects of Perceived Traffic Risks, Noise, and Exhaust Smells on Bicyclist Behaviour: An Economic Evaluation"

_sustainability, doi:10.3390/su11020408_

Reviewer 1 Report

Overall this is a very good paper and examines an area that does not appear to have been studied before. It offers an interesting and significant contribution to the economic evaluation literature around cycling infrastructure.

While I believe there are a few small things and the authors should consider revising

Line 81: This should include disincentives as well as delays

Line 85, 86: While planners and policy makers need new tools and to monetize delay and costs, this paper does not provide a new tool, but rather provide additional knowledge regarding the amount of detours and the costs of these detours. This should be clarified so the authors do not give the misleading impression this is a new tool

Line 100-101: According to the GBD 2016 data, the total for road fatalities was 1.2million of which pedestrians and cyclists make up 27%. However, according to GBD 2016 figures there were 1,342,284 road fatalities, 589077 of which were pedestrians and cyclists, ie 43.9%, substantially higher. Reference should be made to these larger figures to emphasise the importance of road safety for vulnerable road users.

Line 137: The lack of comparison between modes during assessment should be discussed in more detail and emphasised as only assessing one transport mode essentially implies the preferred transport mode has already been chosen, and any CBA undertaken is merely confirming the decision. This comparison goes to the heart of the purpose of the paper as if no comparisons are made, then such issues as noise and smells are never discussed or incorporated into which infrastructure project is most appropriate.

Line 216: It would be good if the authors could explain in more detail how the distances travelled each week were determined. Were these simply estimates from the cyclists themselves or were they given GPS trackers attached to their bikes? If the distances are estimations from the cyclists is there any way to determine a confidence level with them?

Line 217: is the data able to be disaggregated to show shopping trips? With shopping trips would appear to be difficult to avoid smelly, noisy streets, and extra time and effort made to minimise the exposure to such streets may be more pronounced. Either way I think it could add additional insight

Line 219: To be clear, detours were identified specifically for risks, smells or noise? Were these estimates? It is unclear to me if the cyclists could specify whether a detour was made for one specific factor or multi factors. Could this be explained in more detail

Line 286: were the detour %/distances different for leisure versus work? This would imply different value for time for different activities and should be incorporated into the calculations and would seem to be an obvious point of comparison between Figure 2 and Figure 3

Line 394: instead of “probably prohibits a share of cycling” it should be “probably reduces a share of cycling”

Line 398: pollution instead of pollute

Line 401: an issue with this, as found in the planning of the Seville bicycle network, is that the busy streets are the ones with shops and offices and are the places that people want to go. This is why they deliberately chose to put the bike paths on the busy streets as opposed to quieter back roads. See Marqués, R., Hernández-Herrador, V., Calvo-Salazar, M., & García-Cebrián, J. A. (2015). How infrastructure can promote cycling in cities: Lessons from Seville. Research in Transportation Economics53, 31-44.

Line 421: if the restrictions in terms of bicycle network location for some cities such as Seville are correct, then the implications of this study are less for the bicycle infrastructure but more for incentives for electric vehicles that have neither smells and minimal noise and shifts to public transport. These implications should be discussed in more details.

Line 455: Given the implications for electric vehicles and public transport and potential area of research could be to compare the monetary figures per km with various incentives for electric vehicles. This research could explore whether such incentives are aligned and whether the incentives reflect the level of external costs due to noise and smells.

Author Response

Line 81: This should include disincentives as well as delays

We have added "disincentives"

Line 85, 86: While planners and policy makers need new tools and to monetize delay and costs, this paper does not provide a new tool, but rather provide additional knowledge regarding the amount of detours and the costs of these detours. This should be clarified so the authors do not give the misleading impression this is a new tool

The word tool has been replaced with "information"

Line 100-101: According to the GBD 2016 data, the total for road fatalities was 1.2million of which pedestrians and cyclists make up 27%. However, according to GBD 2016 figures there were 1,342,284 road fatalities, 589077 of which were pedestrians and cyclists, ie 43.9%, substantially higher. Reference should be made to these larger figures to emphasise the importance of road safety for vulnerable road users.

We would have liked to include this information, but found it impossible to locate it on the GBD website, in spite of extensive searches.

Line 137: The lack of comparison between modes during assessment should be discussed in more detail and emphasised as only assessing one transport mode essentially implies the preferred transport mode has already been chosen, and any CBA undertaken is merely confirming the decision. This comparison goes to the heart of the purpose of the paper as if no comparisons are made, then such issues as noise and smells are never discussed or incorporated into which infrastructure project is most appropriate.

This is an excellent point which we have included.

Line 216: It would be good if the authors could explain in more detail how the distances travelled each week were determined. Were these simply estimates from the cyclists themselves or were they given GPS trackers attached to their bikes? If the distances are estimations from the cyclists is there any way to determine a confidence level with them?

We have clarified this section in the methodology. As we asked for both distances and travel time, we were able to calculate average cycling speeds. These appear to be realistic at 20 km/h. 

Line 217: is the data able to be disaggregated to show shopping trips? With shopping trips would appear to be difficult to avoid smelly, noisy streets, and extra time and effort made to minimise the exposure to such streets may be more pronounced. Either way I think it could add additional insight

Unfortunately no, we only distinguished work/leisure in this study, to avoid double counting for a wide range of purposes we otherwise would have needed to include. We agree, however, that it would have been desirable to include shopping, though we suspect that for this, mobility diaries would be needed.

Line 219: To be clear, detours were identified specifically for risks, smells or noise? Were these estimates? It is unclear to me if the cyclists could specify whether a detour was made for one specific factor or multi factors. Could this be explained in more detail

Yes, this is correct. First of all, qualitative interviews revealed that risks, smell and noise were not distinguishable for cyclists, i.e. detours had to be defined as multi factor-related. And cyclists had to estimate detours. The sentence now reads: "Estimates of detours as a result of risks, exhaust and noise, i.e. the three factors that together determine the perception of road situations, were recorded in km and minutes of travel. "

Line 286: were the detour %/distances different for leisure versus work? This would imply different value for time for different activities and should be incorporated into the calculations and would seem to be an obvious point of comparison between Figure 2 and Figure 3

Unfortunately, the data does not allow for comparison of detours by purpose (leisure/work), though we agree that this would have been very valuable.

Line 394: instead of “probably prohibits a share of cycling” it should be “probably reduces a share of cycling”

This has been addressed. 

Line 398: pollution instead of pollute

This has been addressed.

Line 401: an issue with this, as found in the planning of the Seville bicycle network, is that the busy streets are the ones with shops and offices and are the places that people want to go. This is why they deliberately chose to put the bike paths on the busy streets as opposed to quieter back roads. See Marqués, R., Hernández-Herrador, V., Calvo-Salazar, M., & García-Cebrián, J. A. (2015). How infrastructure can promote cycling in cities: Lessons from Seville. Research in Transportation Economics53, 31-44.

We agree. The point here is the separation from roads, as many cycle tracks continue to be located on roads. 

Line 421: if the restrictions in terms of bicycle network location for some cities such as Seville are correct, then the implications of this study are less for the bicycle infrastructure but more for incentives for electric vehicles that have neither smells and minimal noise and shifts to public transport. These implications should be discussed in more details.

We believe that this would add a new discussion, which in itself is complex (the policies of urban modal shift change). While we agree that this is a necessary debate, it is perhaps too complex to be added here. 

Line 455: Given the implications for electric vehicles and public transport and potential area of research could be to compare the monetary figures per km with various incentives for electric vehicles. This research could explore whether such incentives are aligned and whether the incentives reflect the level of external costs due to noise and smells.

Again, we would like to abstain from this debate in this paper, as it is perhaps also somewhat premature - electric vehicles are currently very costly, there are issues of private versus external costs, etc.

Reviewer 2 Report

Not much to say on this excellent manuscript, and just the following is suggested:

38 - scooters: does this mode really qualify as active transport?

174 - "given": not clear, please clarify

253 - is 600 km/week reliable?

Table 2: mimimum, Total is 1 km but work and leisure are 0, add decimals to clarify

Table 5: O-barred symbol not clear

Can the conclusions be transferred to non-Northern countries?

References in the text do not meet the journal requirements (full authors name instead of numbers)

Author Response

We are grateful for this reviewer's positive evaluation and the points raised. 

38 - scooters: does this mode really qualify as active transport?

People on scooters may be partially active (kicking off, carrying scooters onto boardwalks, etc.), hence scooters are usually seen as a variant of active transport.

174 - "given": not clear, please clarify 

'given' in the sense of self-evident, indisputable - we have added self-evident to make this clearer.

253 - is 600 km/week reliable?

Reliable no, probable yes - we know cyclists riding over 350 km per week for work, if adding leisure cycling, this is actually quite possible.

Table 2: mimimum, Total is 1 km but work and leisure are 0, add decimals to clarify

We have replaced 0 with<1

Table 5: O-barred symbol not clear

the symbol has been replaced with text, i.e. "average" has been written out

Can the conclusions be transferred to non-Northern countries?

In a general sense, probably, i.e. that exhaust is a nuisance even outside Germany/Austria. 

References in the text do not meet the journal requirements (full authors name instead of numbers)

This has been addressed.

Reviewer 3 Report

Dear Authors, 

I recommend to improve your manuscript in the next points:

- The assertation in lines 41 and 42 "with higher rates in more central and lower incomes areas" should be justified or removed.

- A previous introduction of the acronym "AT" in the table 1 is required.

- To remove the factor "air pollution" in the line 106 in order to a better structured background description about cyclist externalities. 

- The reference to the "snowball sampling; Bryan 2016" is missing (line 163)

- In order to combine the 2 datasets (ADFC and VCÖ) a Wilcoxon test for non-parametric data is more appropriate than a t-test or F-test, due that we are not sure that answers to questionnaires are following a normal distribution.

- The manuscript doesn´t suggest any explanation about the peak in the distances per week: 141-150 and >150 (Figure 2), in the same way that is done for the interval 91-100 Km/week.

-The line 319 should be: "In Denmark, where travel time estimates have been conducted since 2007, the time of cyclist is valued at...."

- References justifying the value of time for cycling are in German (Axhausen et al. 2015), or not available in the referenced web link (Center for transport analytics 2017). The reader doesn´t understand the reason because there is a cost linked to the cyclist time, and why this cost is applied when cycling is leisure related. 

- What is being compared in the Pearson and Spearman tests, and what represent the variables "Noise", "traffic risks" and "Exhaust" in the table 6 is not clear enought for the reader. Are values compared in these tests taken from the survey results under the Likert-scale (used also in the table 4)?

Thanks very much for taking this in consideration.

Best regards.

Author Response

Thank you for these comments that have been very helpful in improving the paper.

- The assertation in lines 41 and 42 "with higher rates in more central and lower incomes areas" should be justified or removed.

The line has been removed.

- A previous introduction of the acronym "AT" in the table 1 is required.

AT has been written out (active transportation)

- To remove the factor "air pollution" in the line 106 in order to a better structured background description about cyclist externalities. 

A good observation that we have considered.

- The reference to the "snowball sampling; Bryan 2016" is missing (line 163)

A reference to Bryman has been added.

- In order to combine the 2 datasets (ADFC and VCÖ) a Wilcoxon test for non-parametric data is more appropriate than a t-test or F-test, due that we are not sure that answers to questionnaires are following a normal distribution.

We have redone the test, now using a non-parametric test (Mann-Whitney-U)

- The manuscript doesn´t suggest any explanation about the peak in the distances per week: 141-150 and >150 (Figure 2), in the same way that is done for the interval 91-100 Km/week.

This has been added

-The line 319 should be: "In Denmark, where travel time estimates have been conducted since 2007, the time of cyclist is valued at....

The sentence has been re-written to be clearer.

- References justifying the value of time for cycling are in German (Axhausen et al. 2015), or not available in the referenced web link (Center for transport analytics 2017). The reader doesn´t understand the reason because there is a cost linked to the cyclist time, and why this cost is applied when cycling is leisure related. 

This has also been adjusted to be clearer.

- What is being compared in the Pearson and Spearman tests, and what represent the variables "Noise", "traffic risks" and "Exhaust" in the table 6 is not clear enought for the reader. Are values compared in these tests taken from the survey results under the Likert-scale (used also in the table 4)?

This has been clarified.

Round  2

Reviewer 3 Report

There are 2 points still unclear:

1.- The penultimate comment:

Willingness-to-pay (WTP) refers to the amount of money cyclists reported to be prepared to pay  for non-exposure to exhaust fumes (lines 190-191)

Travel time is generally considered a private cost which can be measured based on travellers’ willingness to pay (lines 278-280)

In Denmark, where travel time estimates have been conducted since 2007 (Value of time study 2007), the time of cyclists is valued at €12.37/hour (lines 280-281)

Does this means that cyclists are willed to pay 12,37€/km., as a maximum, for using the bicycle instead of other mode of transport, or that cyclists are willed to use the bycicle if they receive 12,37€/km? This continue being a confussion.

2.- The last point: 

What data is considered  for “noise”,”traffic risk” and “exhaust” in the Pearson and Spearman correlation continues without a definitive clarification.

Author Response

Willingness-to-pay (WTP) refers to the amount of money cyclists reported to be prepared to pay  for non-exposure to exhaust fumes (lines 190-191)

Travel time is generally considered a private cost which can be measured based on travellers’ willingness to pay (lines 278-280)

In Denmark, where travel time estimates have been conducted since 2007 (Value of time study 2007), the time of cyclists is valued at €12.37/hour (lines 280-281)

Does this means that cyclists are willed to pay 12,37€/km., as a maximum, for using the bicycle instead of other mode of transport, or that cyclists are willed to use the bycicle if they receive 12,37€/km? This continue being a confussion.

The reviewer seems to misunderstand the concept of travel time value. We have sought to develop the text to better explain what travel time refers to. The section now reads:

These detours represent additional travel time by bicyclists to avoid externalities. Transport planning often seeks to minimize travel time [36,54]. Various studies around the world have monetized the value of travel time and travel time savings for economic evaluation purposes ([4,7,52). Travel time is generally considered a private cost which can be measured based on travellers’ willingness to pay, though this value is context dependent. For example, under favourable conditions, cycling may have a positive value, due to enjoyment and fitness benefits, while under unpleasant conditions, such as wet and cold weather along a busy roadway, costs can be high. In Denmark, where travel time estimates have been conducted since 2007 (Value of time study 2007), the time of cyclists is valued at an average €12.37/hour for unrestricted cycling and €18.69/hour for delays (current estimate according to Center for Transport Analytics [55]). At an average speed of 16 km/h for bicyclists, a travel cost of €0.790/km is the current basis (in 2017) for calculations in Denmark. Far lower values have been reported for Germany, where [48] suggest a value of time of €4.66/hour, resulting in a cost of about €0.3/km for cyclists (without consideration of leisure/work trip purposes). Considering this latter value, the cost of time “lost” to detours of 7.24 km/week, multiplied with €0.30, i.e. about €2.17/week for the average cyclist in this survey, and referring to average cycling conditions. Note that this cost involves traffic risks, exhaust and noise.

What data is considered  for “noise”,”traffic risk” and “exhaust” in the Pearson and Spearman correlation continues without a definitive clarification.

We have added an explanation, and the section now reads:

To understand the statistical dependence between WTA/WTP and traffic risks in comparison to exhaust and noise, Pearson and Spearman rank correlation coefficients were determined (Table 6). The calculation is based on metric WTA/WTP values and ordinal Likert-scale values for noise, traffic risks and exhaust. WTA correlations are negative as expected, and significant at α=0.05 for noise (Pearson and Spearman) and traffic risks (Pearson), as well as significant at α=0.01 for exhaust (Pearson and Spearman). This highlights the importance of exhaust as a barrier effect that has so far not received much attention as a negative externality in transport CBA.